# The Soluble Cytoplasmic Tail of CD45 (ct-CD45) Regulates Dendritic Cell Activation and Function via TLR4 Signaling

**DOI:** 10.3390/ijms26083888

**Published:** 2025-04-20

**Authors:** Sara Gil-Cantero, Alexander Puck, Sarojinidevi Künig, Veronica Pinnarò, Petra Waidhofer-Söllner, Johannes Stöckl

**Affiliations:** Institute of Immunology, Center of Pathophysiology, Infectiology and Immunology, Medical University of Vienna, 1090 Wien, Austria; sara.gilcantero@meduniwien.ac.at (S.G.-C.); veronica.pinnaro@meduniwien.ac.at (V.P.); petra.waidhofer-soellner@meduniwien.ac.at (P.W.-S.)

**Keywords:** CD45, TLR4 signaling, dendritic cells, immune regulation

## Abstract

The soluble cytoplasmic tail of the prototypic receptor-like protein tyrosine phosphatase (PTP) CD45 (ct-CD45) is cleaved and released into the human plasma by activated phagocytes. Released ct-CD45 was found to inhibit T cell proliferation and cytokine production via engagement of Toll-like receptor 4 (TLR4). In this study, we analyzed the impact of the ct-CD45/TLR4 pathway on the function of human monocyte-derived dendritic cells (DCs). We could demonstrate that activation of DCs by ct-CD45 upregulated the expression of certain cell surface markers (e.g., CD71 and CD86) and induced IL-10 production via TLR4. Yet, in contrast to stimulation with LPS, other typical cell surface markers and cytokines were not upregulated or induced in DCs by ct-CD45. The T cell proliferation–stimulatory capacity of DCs was not modulated by ct-CD45 treatment. However, treatment of DCs with ct-CD45 modulated the cytokine profile in co-cultured T cells. While IFN-γ production induced by DCs was strongly inhibited, the release of IL-4 was increased in T cells upon stimulation with ct-CD45-treated DCs. In contrast, ct-CD45-stimulated DCs induced IL-2 and IL-10 production in co-cultured T cells comparable to untreated DCs. In summary, we could demonstrate that ct-CD45 acts as an immunoregulatory factor for DCs via a non-canonical TLR4-dependent activation pathway.

## 1. Introduction

Numerous cell surface receptors of immune cells are subject to regulated intramembrane proteolysis (RIP), a highly conserved cellular process that can generate soluble protein fragments with new biological activity [1]. Our group has identified CD45, which is the prototypic receptor-like protein tyrosine phosphatase expressed on leukocytes, as a target for RIP during the activation of human phagocytes [2,3]. This initial proteolytic event is succeeded by γ-secretase-mediated cleavage and release of the CD45 intracellular domain. This soluble fragment termed the “cytoplasmic tail of CD45” (ct-CD45) is 95 kDa in size and contains both phosphatase homology domains. After its processing, ct-CD45 is released from dying phagocytes and was found to act as a cytokine-like factor on human T lymphocytes that potently inhibits their proliferation independent of the protein’s phosphatase activity [3]. Further studies then indicated that ct-CD45 acts as an inhibitor of T cell activation induced by dendritic cells (DCs) or via plate-bound antibodies [3,4,5]. However, ct-CD45 is not a mere inducer of T cell anergy but rather an extrinsic regulator of T cell activation thresholds since it preferentially suppresses T cell activation induced via CD3/TCR (signal 1) but not in the presence of co-stimulation via CD28 (signal 2) [4]. This hypothesis is supported by the finding that ct-CD45-treated T cells have gene expression profiles that are also found in quiescent lymphocytes, such as low expression of cyclins, but they have high levels of the CDK inhibitor p27kip1 or the putative quiescence factors KLF2 and SLFN12 [4].

Subsequent studies demonstrated that ct-CD45 binds to Toll-like receptor 4 (TLR4) and acts via a non-canonical TLR4 activation pathway on T cells, which modulates TCR signaling.

TLR4 is a pattern recognition receptor that has been characterized to bind the bacterial cell wall component lipopolysaccharide [5]. However, during the last years, numerous endogenous ligands have been described for TLR4, including, for example, proteins like High-Mobility Group Protein B1 (HMGB1), heat shock proteins, and tenascin-c but also phospholipids like 1-palmitoyl-2-arachidonyl-sn-glycero-3-phosphorylcholine (oxPAPC) [6,7,8,9,10]. Endogenous TLR ligands are derived from damaged host tissues or activated cells, subsequently binding to immune cells to initiate downstream signaling [6,9]. Depending on the cellular context and the underlying disease, these signals may be pro-inflammatory and favor immune cell recruitment to the site of infection; however, they may also promote tissue repair and wound healing or even induce inhibitory effects in affected tissues [11,12,13].

DCs play a crucial role in sensing and responding to endogenous or exogenous danger signals in the body [14,15,16]. They act as sentinels of the immune system, constantly surveying their surroundings for signs of infection, tissue damage, or other threats. TLR4 is one of the central receptors involved in this guarding function. Since ct-CD45 is a component of human plasma and appears to contribute to its immunoregulatory characteristics, we analyzed the direct impact of ct-CD45 on DC function, in particular their T cell stimulatory capacity, in this study. The results of this study demonstrate that ct-CD45 acts as an immunoregulatory factor on DCs via non-canonical TLR4 activation, which alters the polarization of co-cultured T cells.

## 2. Results

### 2.1. Immature DCs Upregulate CD86 and CD71 upon ct-CD45 Stimulation

TLRs play important roles in the sensing of infection by innate immune cells. Professional antigen-presenting cells (APCs), such as DCs, are well equipped with these receptors. TLR signaling induces the functional maturation of these cells, which is characterized by the upregulation of MHC molecules, costimulatory receptors, and the induction of cytokine production. Thus, we reasoned that ct-CD45 might also influence the function of DCs. To test this hypothesis, highly purified monocytes were differentiated into immature DCs and then stimulated with ct-CD45; half of the cells were matured by the addition of LPS for 2 days. Subsequently, the cells were analyzed via flow cytometry for the expression of maturation markers on the cell surface. As expected, LPS stimulation of DC upregulated the expression of the costimulatory molecules CD86 and CD40 as well as the expression of MHC class II (Figure 1A). It also promoted the induction of the transferrin receptor (CD71) and the coinhibitory ligand for PD-1 (B7-H1, PD-L1, CD274), while the phagocytic receptor DC-SIGN (CD209) and the characteristic marker for DC (CD1a) were slightly downmodulated. The expression of CD74, the MHC class II invariant chain, was not significantly altered by LPS or ct-CD45 treatment alone. In comparison, treatment of immature DCs with recombinant ct-CD45 also leads to a significant upregulation of CD86 and CD71 (Figure 1B,C). Other cell surface markers were not significantly altered by ct-CD45 treatment, although a tendency for CD209 downregulation was observed (Figure 1B,C). Co-treatment of DCs with LPS and ct-CD45 led to phenotypically normal DC maturation since no significant differences were observed compared to cells stimulated with LPS alone (Figure 1B,C).

### 2.2. ct-CD45-Stimulated DC Release Increased Levels of IL-10

Activation of DCs is known to release a plethora of soluble mediators, including cytokines and chemokines. The upregulation of CD86 and CD71 on DCd upon ct-CD45 treatment suggested a partially activated phenotype. Hypothesizing that these changes were likely to be reflected by increased cytokine production of ct-CD45 treated DCs, we collected supernatants of these cultures and analyzed the release of five typical cytokines, comprising IL-10, IL-12, IL-6, IL-23, and TNF-α.

The results presented in Figure 2 demonstrate that the treatment of DCs with LPS strongly stimulated the production of all five cytokines analyzed. Conversely, the administration of ct-CD45 predominantly induced IL-10 production in immature DCs (Figure 2). In contrast to LPS, ct-CD45 treatment had no significant effect on the release of IL-12, IL-23, TNF-α, or IL-6 by DCs, indicating a polarization of the cells towards an anti-inflammatory cytokine profile.

### 2.3. ct-CD45 Binds to Human DCs in a Dose-Dependent Manner

Ct-CD45 displays distinct binding to primary human T cells, which can be interfered with by a monoclonal antibody directed against the CD45 intracellular domain. Since we observed the functional effects of ct-CD45 treatment also for human DCs, we aimed to analyze the direct binding of ct-CD45 to these cells.

Thus, DCs were incubated with graded amounts of ct-CD45. Recombinant His-tagged ct-CD45 protein showed dose-dependent binding to both immature and LPS-matured DCs, with maximum binding observed at a concentration of 80 µg/mL (Figure 3). So, in contrast to T cells, the activation of DCs is not required to promote the binding of ct-CD45.

### 2.4. ct-CD45-Mediated the Increase in DC Surface Markers, and Cytokine Production Can Be Inhibited by a TLR4 Blockade

Since ct-CD45 effects on primary human T cells could be inhibited via a TLR4 blockade, we hypothesized that the effects observed for DCs were also TLR4 dependent. To this end, DCs were stimulated with ct-CD45 and co-incubated with anti-TLR4 mAb W7C11 or the TLR4 signaling inhibitor CLI-095. As before, cell culture supernatants were analyzed for IL-10 cytokine secretion, and cells were stained for the activation marker CD71.

The ct-CD45-mediated upregulation of IL-10 secretion in immature DCs could be significantly antagonized by both inhibitors that were tested (Figure 4A). Also, ct-CD45-mediated induction of the cell surface receptors CD71 could have significantly interfered with the blockade of TLR4 with anti-TLR4 mAb W7C11 or the TLR4 signaling inhibitor CLI-095 (Figure 4B). Like with ct-CD45 stimulation, the activation of DCs with LPS in the presence of the TLR4 antagonism reduced IL-10 release and CD71 expression (Figure 4A,B). Taken together, targeting of TLR4 signaling inhibits ct-CD45-mediated DC activation and polarization.

### 2.5. ct-CD45 Treatment of DCs Does Not Affect the Proliferation of Co-Cultured T Cells

The induction of T cell proliferation is a hallmark of DCs, in particular mature DCs. So, we next analyzed if ct-CD45 treatment affects the T cell stimulatory capacity of DCs. The results presented in Figure 5 demonstrate that treatment of immature DCs with ct-CD45 had no impact on the induction of proliferation of allogeneic T cells upon co-culture, independent of the DC:T ratios used in this study. In contrast, T cell proliferation was increased upon co-culture with LPS-stimulated DCs (Figure 5).

### 2.6. Treatment of DCs with ct-CD45 Favors the Induction of Th2-Type T Cell Responses

Next, we investigated whether ct-CD45 treatment of DCs might affect the functional polarization of co-cultured T cells. The results presented in Figure 6 demonstrate that ct-CD45 treated DCs induced less IFN-γ production in co-cultured T cells but higher amounts of IL-4. We also observed an increase in GM-CSF production in T cells stimulated with ct-CD45-treated DCs, but the effect did not reach statistical significance. Surprisingly, IL-10 production was rather reduced in T cells stimulated with ct-CD45-treated DCs compared to untreated DCs. IL-2 production was not significantly altered in T cells stimulated with immature DCs activated by ct-CD45. Thus, ct-CD45-stimulated DCs seemingly promote the induction of a Th2-type response in co-cultured T cells.

## 3. Discussion

Dendritic cells (DCs) are key sentinels of our body against invading pathogens for the initiation of immediate innate immune reactions and subsequent translation of this information into appropriate adaptive immune responses [17,18]. In this study, we analyzed the impact of ct-CD45/TLR4 interactions on DC function. TLRs are well known for their recognition of conserved PAMPs, which is the prerequisite for the activation of innate immune cells and the ensuing induction of adaptive immunity towards a pathogen [10]. However, an increasing number of endogenous TLR ligands have been identified, which can provide a mechanistic explanation for the induction and maintenance of sterile inflammatory responses, such as systemic inflammatory response syndrome or autoimmune disorders, like rheumatoid arthritis (RA) [9]. The results of this study demonstrate that engagement of TLR4 through ct-CD45 induces a non-canonical activation program in DCs. As a consequence, the function of TLR4 on DCs is altered, and the induction of Th2-type T cell responses is favored by these pivotal antigen-presenting cells.

Ct-CD45 is a component of human plasma that is reduced in patients with autoimmune diseases [4]. Recent studies have further demonstrated that ct-CD45 binds to TLR4 and signals via TLR4 to inhibit T cell activation via TCR [5]. Engagement of TLR4 by ct-CD45 activates NFκB, a classical pathway to induce functional maturation of DCs [5,19]. Intriguingly, unlike in T cells, ct-CD45 promoted the activation of immature DCs since it induced the expression of CD71 and CD86 on the cell surface but had no apparent effect on the marker expression of LPS-matured cells. However, DCs stimulated via ct-CD45 displayed a polarized pattern of cytokine release, characterized by the upregulation of IL-10 but not IL-12 or IL-23. While IL-10 is widely recognized for its immunoregulatory functions, IL-12 and IL-23 are rather pro-inflammatory cytokines that promote Th1- and Th17-type T cell responses, respectively. Like for T cells, the blockade of TLR4 with mAb W7C11 or the specific inhibitor CLI-905 was able to antagonize ct-CD45-mediated effects, suggesting signal transduction via this receptor. Taken together, these results demonstrate that the engagement of DCs with ct-CD45 leads to partial activation of the cells concerning the expression of activation markers of DCs as well as the induction of pro- and anti-inflammatory cytokines via a TLR4-dependent signaling pathway.

TLR4 has a dual signaling potential via MyD88- and TRIFF-dependent routes [11,20]. Yet, the activation profile of ct-CD45 was significantly different compared to DCs activated via LPS, the classical pathogen-derived ligand and agonist of TLR4, concerning the cytokine release and upregulation of DC characteristic activation markers (Figure 1 and Figure 2). Stimulation with ct-CD45 and LPS did not synergistically increase or inhibit the expression of cell surface markers. Thus, ct-CD45 is seemingly not a simple inhibitor of LPS-induced maturation of DCs but is modulating the function of TLR4 and, consequently, the activation of DCs. LPS-mediated signaling via TLR4 involves several accessory proteins to form the TLR4 receptor complex [10,21,22]. LPS is initially bound in plasma by the LPS-binding protein (LBP) and then transferred to CD14, which exists in soluble form as well as membrane bound via glycosylphosphatidylinositol. Both LPB and CD14 are required to enhance the sensitivity of TLR4 towards LPS. CD14 is also necessary for the endocytosis of TLR4 upon LPS binding and the initiation of TRIF-dependent signaling. CD14 transfers LPS to the TLR4-associated protein MD-2, and signaling is initiated by the homodimerization of LPS/TLR4/MD-2 complexes [10,20]. Thus, it is possible that ct-CD45 interacts with the TLR4 complex and acts as another accessory molecule which contributes to the fine tuning of the downstream signals and subsequent functional effects.

In addition, TLR4 signaling might be even more complex than outlined above and involve additional interaction partners to enable signaling via different endogenous ligands. Recent studies suggest that TLR4 signaling involves the recruitment of TLR4 and other accessory proteins, including, for example, CD36 or CD44, to lipid rafts [23]. In light of these findings, it appears plausible to suggest the interaction of ct-CD45 with receptors engaged in lipid raft-mediated TLR4 signaling platforms. Moreover, ct-CD45 might have parallels to soluble CD83, which was described to alter TLR4 signaling on monocytes by binding to the TLR4 co-receptor MD-2 [24]. Endocytosis is another important mechanism that regulates the signaling pathways induced via TLR4 [10]. Since ct-CD45 has two potential binding sites for TLR4, it is intriguing to speculate that binding to ct-CD45 might be able to cross-link and reorganize TLR4 on the surface of DCs, which might subsequently influence its functional behavior and capacity when LPS triggers such TLR4. So, ct-CD45 can be seen as a molecular brick of this complex unit forming the TLR4 platform, which is used by our immune system to modulate the function of DCs.

The engagement of TLR4 by exogenous and endogenous ligands is well established for the induction of inflammatory immune responses [21]. However, not all endogenous TLR4 ligands promote inflammation. For instance, oxidized phospholipids, such as oxPAPC, are endogenous antagonists of both TLR2 and TLR4 that appear to compete with TLR accessory proteins, which interact with the microbial ligands of these receptors [7,25]. Nevertheless, even agonistic signaling via TLR4 can be diverted, leading to different functional outcomes. Prolonged stimulation with LPS is known to result in a state of tolerance, which is characterized by reduced expression of inflammatory cytokines both in vitro and in vivo [6,26]. This effect is dose dependent and can be enhanced upon repeated administration of LPS. Furthermore, Piccinini and colleagues demonstrated that different TLR4 ligands may induce divergent signaling pathways, leading to the expression of both overlapping and distinct gene products [27]. In that study, it was found that the infectious stimulus LPS promotes the expression of extracellular matrix-degrading proteases, like MMP13, by human macrophages. Conversely, the endogenous TLR4 ligand tenascin-C, which is released upon tissue damage, induces the synthesis of matrix-forming molecules, including type I collagen, in these cells. Both ligands are pro-inflammatory since they share the induction of the cytokines IL-6 and IL-8, although only LPS stimulates the synthesis of IL-23 and IL-12 in macrophages [27]. Yet, the mechanism of how ct-CD45 modulates the function of TLR4 on DCs needs to be clarified in future studies.

DCs are famous for their T cell stimulatory capacity [18]. The results of our study demonstrate that ct-CD45 activation of DCs is not lifting or inhibiting the capacity of DCs to induce proliferation in co-cultured T cells. Yet, we observed that functional polarization of T cells stimulated with ct-CD45-treated DCs was significantly shifted towards Th2-type cytokine production. In contrast, induction of the prototypic Th1-type cytokine IFN-γ was inhibited by ct-CD45 DCs. Ct-CD45 is present at high levels in human plasma during the steady state but is significantly reduced in plasma derived from patients suffering from Systemic Lupus Erythematosus (SLE) or RA, which suggests that ct-CD45 contributes to the pathogenesis of these autoimmune diseases [4]. Our present results indicate that ct-CD45 is a TLR4 agonist that potentially diverts TLR4 signaling pathways towards the expression of immunoregulatory mediators. Thus, it is tempting to suggest that RA or SLE patients might benefit from the exogenous administration of ct-CD45 as a therapeutic drug. In summary, we could demonstrate that ct-CD45 acts as an immunoregulatory factor on human DCs via non-canonical TLR4 activation.

## 4. Materials and Methods

### 4.1. Media, Reagents and Chemicals

Cells were maintained in RPMI 1640 medium (Sigma, St. Louis, MO, USA) supplemented with 2 mmol/L L-glutamine, 100 U/mL penicillin, 100 μg/mL streptomycin, and 10% fetal calf serum (Gibco, Grand Island, NY, USA). Lipopolysaccharide (LPS) from Escherichia coli was also obtained from Sigma-Aldrich. PBS was obtained from Lonza (Verviers, Belgium).

The following murine monoclonal antibodies (mAbs) were raised in our laboratory: negative control antibody VIAP (against calf intestine alkaline phosphatase), CD14 (MEM18), CD1a (VIT6b), CD209, CD71 (5-528), MHC-II (VID1), CD74, and CD274 (PD-L1, B7-H1 mAb 5-272). mAb CD284 (TLR4) was purchased and obtained from Invitrogen (IL, USA). Anti-hTLR4-IgG (W7C11) and CLI-095 (Resatorvid) were obtained from Invivogen. Hybridoma-producing mAb G28-5 (CD40) was obtained from the American Tissue Culture Collection (ATCC). mAb CD86 (BU63) was supplied by Thermo Fisher Scientific Inc. (Paisly, UK). Protein tyrosine phosphatase CD45 catalytic domain a.a. 632-1304 with N-terminal His-tag was acquired from BPS Bioscience (San Diego, CA, USA). The endotoxin levels of all lots of ct-CD45 used in this study were <1 EU/mL. Human recombinant GM-CSF and IL-4 were provided by Novo Nordisk A/S (Bagsværd, Denmark).

### 4.2. Generation of Monocyte-Derived DCs

Human monocytes were isolated from anonymized Buffy Coats which were bought from the Austrian Red Cross. According to Austrian law and the assessment of the ethical committee, the use of anonymized, not identifiable cells obtained from a commercial provider does not require specific ethical approval. This position is based on the Preamble of “WMA DECLARATION OF HELSINKI—ETHICAL PRINCIPLES FOR MEDICAL RESEARCH INVOLVING HUMAN SUBJECTS”.

Peripheral blood mononuclear cells (PBMCs) were isolated via standard density gradient centrifugation using Ficoll-Paque TM Plus (GE Healthcare, Amersham, UK), and monocytes were purified from PBMCs using the MACS system (Miltenyi Biotec, Bergisch Gladbach, Germany) as described previously [25,28]. CD14+ monocytes were cultured for 6 days in RPMI 1640 culture medium supplemented with 0.5% L-glutamine, 1% Penicillin/Streptomycin (PAA Laboratories, Pasching, Austria), and 10% heat-inactivated fetal calf serum. For triggering differentiation to monocyte-derived dendritic cells (DCs), human recombinant cytokines were used at the following concentrations: 50 ng/mL − GM-CSF + 200 U/mL IL-4. For the treatment with ct-CD45, the protein was used at a final concentration of 30 µg/mL, like in previous studies [3]. In order to block the TLR4 pathway, DCs were pre-treated with mAb W7C11 (final concentration 5 µg/mL) or CLI-095 (final concentration 1 µg/mL) at 37 °C for 90 min before adding ct-CD45 and/or LPS to the cells.

### 4.3. Mixed Leukocyte Reaction

For mixed leukocyte reaction (MLR), purified T cells (1 × 10^5^ cells/well) were stimulated with allogeneic DCs (2 × 10^4^ cells/well), if not otherwise stated. Experiments were performed in 96-well round-bottom cell culture plates in the presence of RPMI-1640 medium (Mock) or indicated cell supernatants, as described previously. T cell proliferation was monitored, measuring [methyl-3H]thymidine incorporation on day 5. Cells were harvested 18 h later, and radioactivity was determined on a microplate scintillation counter (PerkinElmer Life and Analytical Sciences, Waltham, MA, USA). Assays were performed in triplicate.

### 4.4. Surface Marker Staining of Dendritic Cells

DCs were harvested, and supernatants were kept for cytokine analysis. A total of 1 × 10^5^ cells per condition were separated, and pellets were resuspended in Beriglobin for Fc blocking. A 50 µL cell suspension was then stained with primary mAb for 60 min at 4 °C. After the primary staining step, cells were washed two times and stained with a secondary antibody labeled with Oregon Green 488 for 30 min at 4 °C. DCs were then washed again 2× and resuspended in 50 µL flow cytometry fluid for measurement of surface marker expression. Flow cytometry analyses were performed using the flow cytometer Calibur (Becton Dickinson, Fanklin Lakes, NJ, USA).

### 4.5. ct-CD45 Binding to DCs

For analysis of ct-CD45 binding to cells, recombinant ct-CD45 protein (His tagged) was used instead of primary mAb at the indicated concentrations (20, 40, and 80 µg/mL) and incubated for 60 min at 4 °C. After washing, bound protein was detected using Alexa Fluor 647-conjugated anti-6x His-tag mAb (Invitrogen, Carlsbad, CA, USA). After washing, cells were resuspended in FACS sheath fluid until analysis on a FACS Calibur flow cytometer.

### 4.6. Cytokine Measurements

Supernatants from DC experiments and T cells co-cultured with DCs were harvested and stored at −20 °C for further analysis. Cytokines including IL-6, IL-10, IL-12, IL-23, TNF-α, IL-2, IL-4, GM-CSF, and IFN-γ were measured using the MILLIPLEX Map Kit (Merck, Darmstadt, Germany) as described in the manufacturer’s protocol. All measurements were performed in triplicates.

### 4.7. Statistical Analysis

Data were analyzed using GraphPad Prism 9.5.1 software (GraphPad Software, La Jolla, CA, USA). For the surface marker expression of DCs, paired *t*-tests were applied, while for the other experiments, one-way ANOVA and post hoc pairwise Tukey’s tests were applied. For T cell proliferation, a two-way ANOVA test followed by a post hoc pairwise Tukey’s test was applied. *p* values were * *p* < 0.05; ** *p* < 0.01; and *** *p* < 0.001.

## Figures and Tables

**Figure 1 ijms-26-03888-f001:**
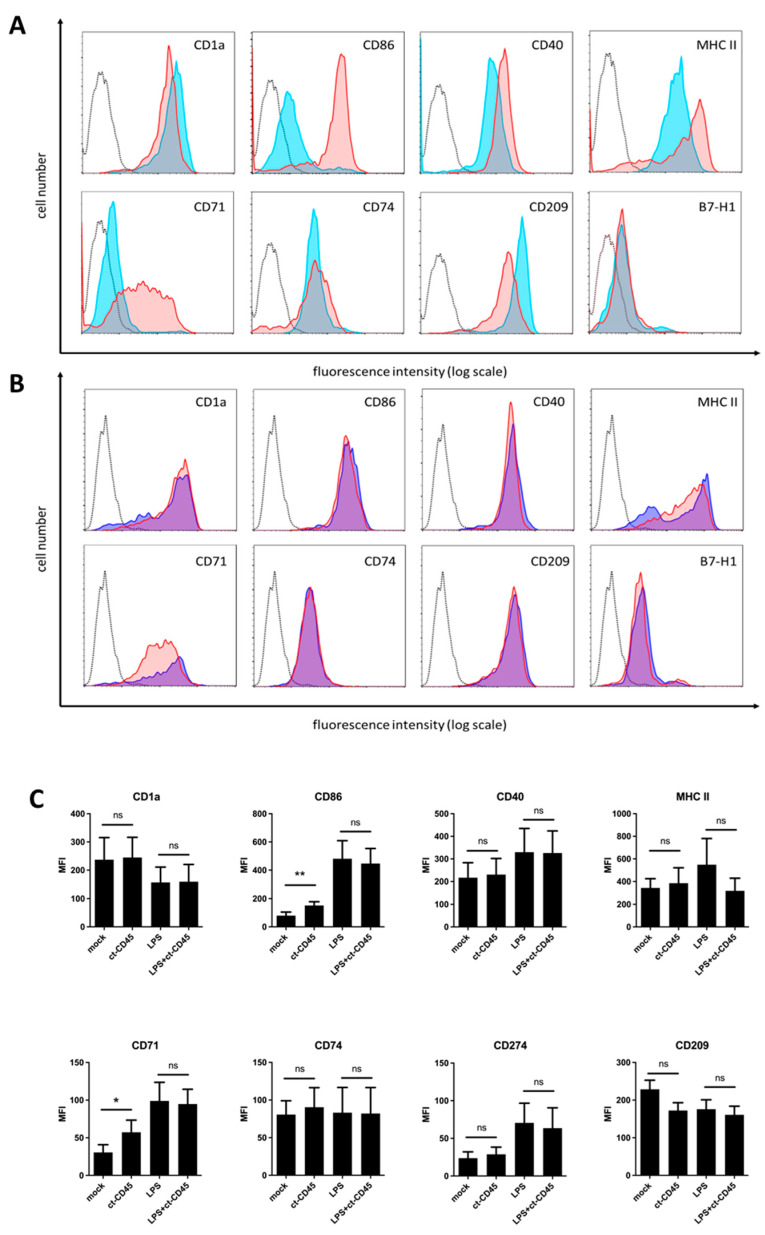
Immature monocyte-derived dendritic cells (DCs) upregulate CD86 and CD71 upon ct-CD45 stimulation. (**A**) Primary human monocytes (CD14+) were differentiated into immature DCs using IL-4 and GM-CSF. On day 5 of differentiation, the cells were treated with ct-CD45 (30 µg/mL) or left untreated. Cells were then (**A**) left untreated or stimulated with immobilized ct-CD45 for 2 days. (**B**) Immature DCs were stimulated for 2 days with LPS (1 µg/mL) or a combination of LPS and ct-CD45. Two days after stimulation, the cells were harvested, and the indicated markers were stained and assessed via flow cytometry. (**A**–**C**) Flow cytometry plots from representative experiments are shown: black dotted line (isotype control staining), light blue histogram (untreated cells), dark blue (LPS treatment), red (ct-CD45 treatment) without (**A**) or with LPS co-treatment (**B**). (**C**) Data from 5 independent experiments using 7 different donors were pooled and displayed as mean ± SEM (*n* = 7). Student’s paired *t*-test with * *p* < 0.05, ** *p* < 0.01 and ns (not significant).

**Figure 2 ijms-26-03888-f002:**
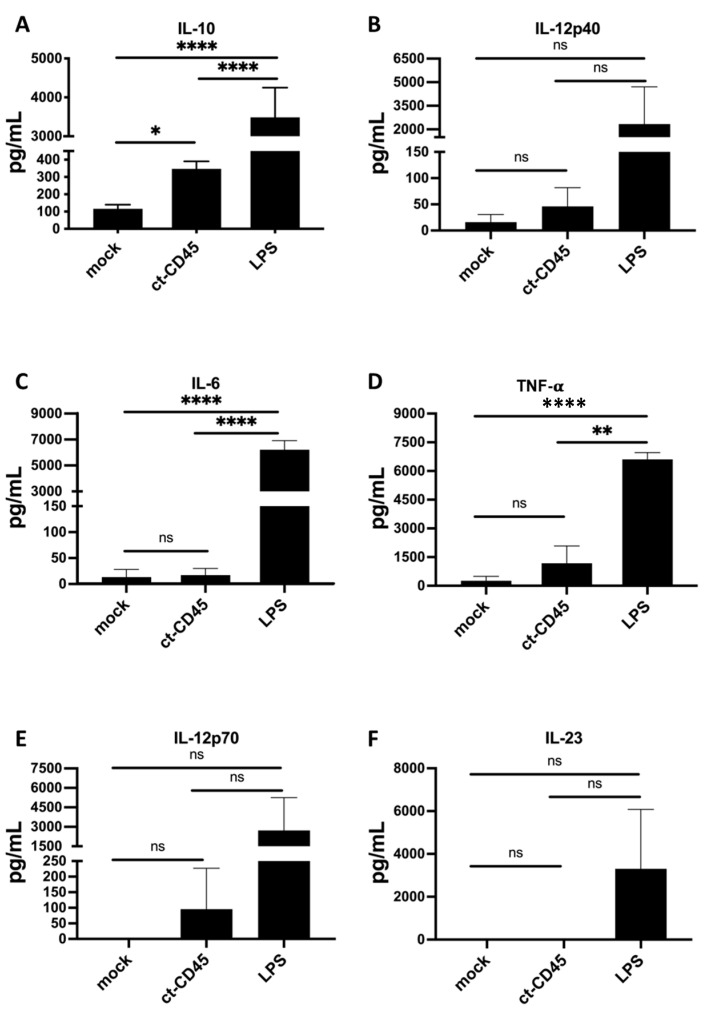
ct-CD45 stimulates DCs to produce IL-10. Immature DCs were stimulated with LPS (1 µg/mL) or ct-CD45 (30 µg/mL) or were left untreated and cultivated for another 2 days. Cell culture supernatants were harvested, and IL-6, IL-10, IL-12, IL-23, and TNF-α levels were measured via MILLIPLEX multiplex assays using the Luminex system (**A**–**F**). Data displays mean ± SEM of pooled data from 5 independent experiments using 7 donors in total (*n* = 7). Statistical analysis was performed via one-way ANOVA followed by post hoc pairwise Tukey’s tests with * *p* < 0.05, ** *p* < 0.01, **** *p* < 0.0001 and ns (not significant).

**Figure 3 ijms-26-03888-f003:**
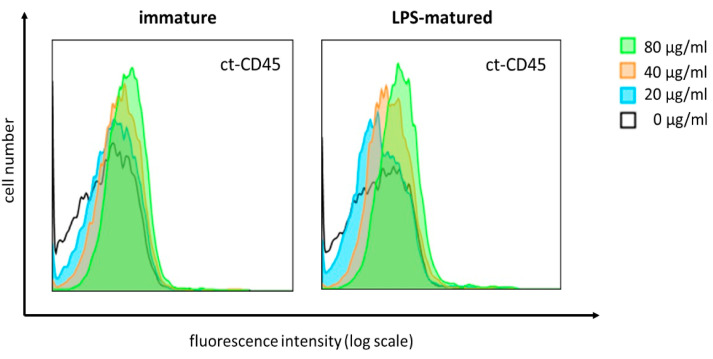
ct-CD45 binds to human DCs in a dose-dependent manner. After day 5 of in vitro generation from monocytes, human DCs were either left untreated or stimulated with LPS (1 µg/mL). Two days after stimulation, DC were harvested and incubated with the indicated concentrations of recombinant ct-CD45 followed by detection with Alexa Fluor 647-labelled anti-6x-His mAb. Flow cytometry plots representative of 2 independent experiments are shown.

**Figure 4 ijms-26-03888-f004:**
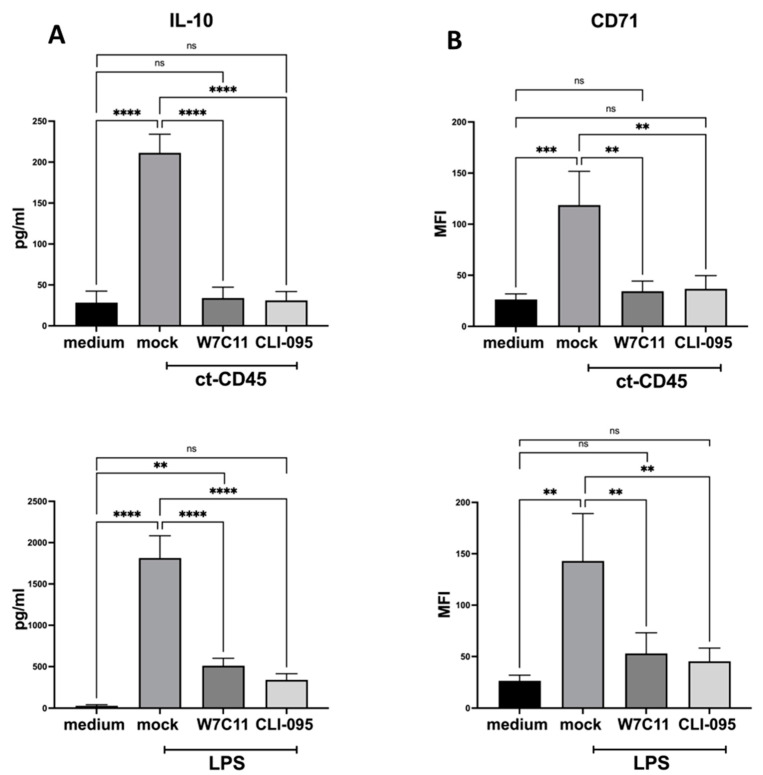
ct-CD45-mediated cytokine production in DCs can be inhibited by a TLR4 blockade. Human DCs were cultivated in IL-4/GM-CSF-containing medium and were stimulated for 2 days with ct-CD45 or LPS and were treated with either TLR4-neutralizing mAb (W7C11) at 5 µg/mL or CLI-095 (100 ng/mL). Subsequently, the cultures were harvested, supernatants were analyzed for IL-10 production (**A**), and (**B**) DCs were analyzed for CD71 cell surface expression via flow cytometry. Mean ± SEM of 3 independent experiments is displayed. Statistical analysis was performed via one-way ANOVA followed by post hoc pairwise Tukey’s tests with ** *p* < 0.01, *** *p* < 0.001, **** *p* < 0.0001 and ns (not significant).

**Figure 5 ijms-26-03888-f005:**
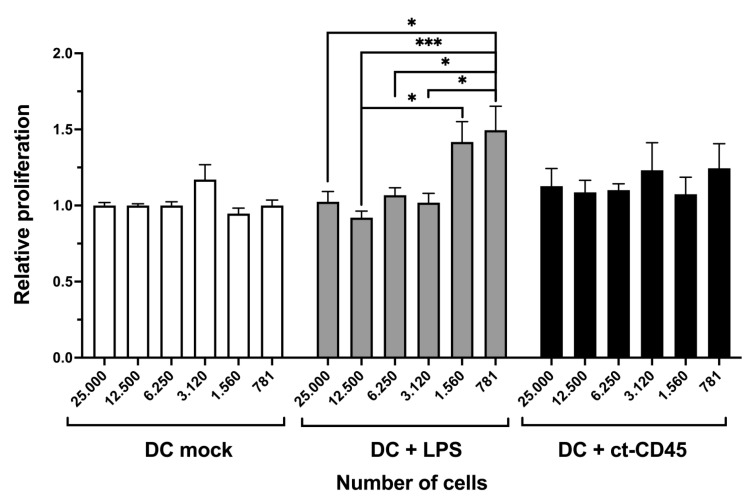
ct-CD45 enhances the stimulatory capacity of mature DCs to induce T cell proliferation. Human DCs were cultivated in IL-4/GM-CSF-containing medium and then stimulated for 2 days via ct-CD45 or LPS. DCs were then co-cultured with allogeneic T cells, and proliferation was analyzed by H3-thymidine incorporation and measured on a beta counter. Mean ± SEM of 8 independent experiments is displayed. Statistical analysis was performed via two-way ANOVA followed by post hoc pairwise Tukey’s tests with * *p* < 0.05, *** *p* < 0.001.

**Figure 6 ijms-26-03888-f006:**
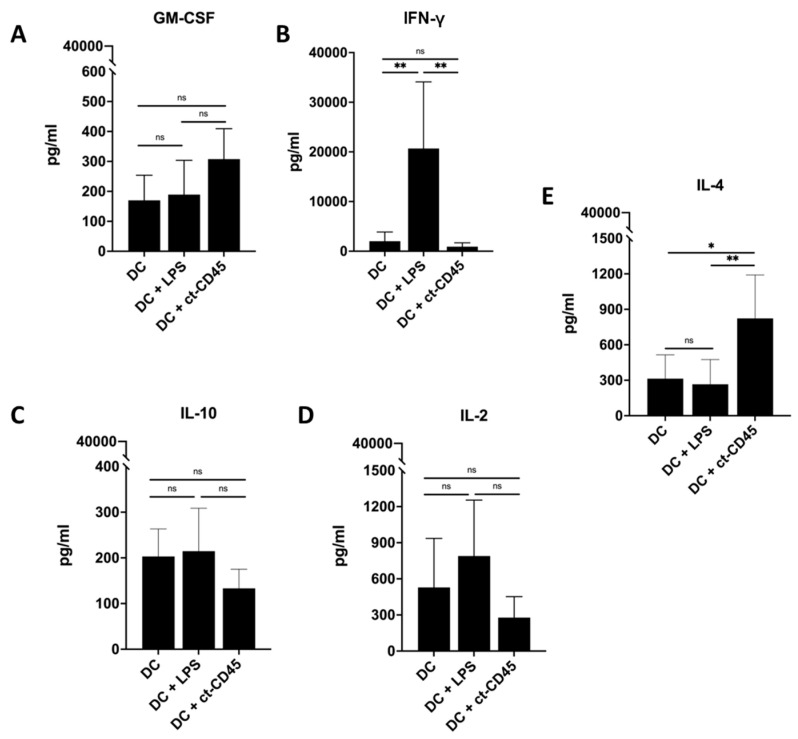
ct-CD45 enhances the stimulatory capacity of DCs to induce type 2 T cell polarization. Production of cytokines by T cells was analyzed after 3 days of co-culture with untreated (mock DC), LPS-stimulated, or ct-CD45-stimulated DCs. (**A**) GM-CSF, (**B**) IFN-γ, (**C**) IL-10, (**D**) IL-2, (**E**) IL-4. *n* = 6. Statistical analysis was performed via one-way ANOVA followed by post hoc pairwise Tukey’s tests with * *p* < 0.05, ** *p* < 0.001 and ns (not significant).

## Data Availability

The raw data of this study are available upon request from the corresponding author.

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
