# Peer review of "The Soluble Cytoplasmic Tail of CD45 (ct-CD45) Regulates Dendritic Cell Activation and Function via TLR4 Signaling"

_ijms, 2025, doi:10.3390/ijms26083888_

Round 1

Reviewer 1 Report

Comments and Suggestions for Authors

The authors examine signaling of the soluble tail of CD45 (ct-CD45) through TLR4 in human monocyte-derived dendritic cells. The phenotypic and functional consequences of ct-CD45 mediated TLR4 activation in dendritic cells are reported.

The authors have originally described this unique signaling pathway in T cells and now extend their research to dendritic cells. Overall, this is solid and appealing work.

I wonder whether ct-CD45 activated, CD71-expressing DCs exhibit regulatory activity by competing for iron during T cell activation (CD71 blockade?). And along the same line: are the induced T cells anergic or can they be reactivated? 

The authors should point out that CLI-095 is also known as resatorvid. I think the discussion is a little too long and could be shortened.

Author Response

Thank you very much for your interest in our study and your constructive comments and feedback. We would like to address the comments and questions in the following way:

Reviewer: The authors examine signaling of the soluble tail of CD45 (ct-CD45) through TLR4 in human monocyte-derived dendritic cells. The phenotypic and functional consequences of ct-CD45 mediated TLR4 activation in dendritic cells are reported.

The authors have originally described this unique signaling pathway in T cells and now extend their research to dendritic cells. Overall, this is solid and appealing work.

I wonder whether ct-CD45 activated, CD71-expressing DCs exhibit regulatory activity by competing for iron during T cell activation (CD71 blockade?). And along the same line: are the induced T cells anergic or can they be reactivated? 

Response: Thank you for this interesting question. You are right that iron plays an important role for T cell biology and in particular proliferation. Iron deficiency causes an almost complete inhibition of proliferation and a deep anergic state. Yet, we have not observed that ct-CD45 stimulated DCs affect the proliferation of co-cultured T cells (Figure 5). So, it is likely that there is enough iron available for the cells in spite of the higher CD71 expression on ct-CD45-treated DCs. Moreover, we have recently demonstrated that iron depletion is not a polarizing process for T cells since the cells produce normal levels of IL-2, IL-4 and IFNg (Berg et al. ImmunoHorizons, 2020, 4:165). So, it seems to be unlikely that CD71 on DCs might be important in our setting for the Th2 differentiation which we found in this study.

Reviewer: The authors should point out that CLI-095 is also known as resatorvid. I think the discussion is a little too long and could be shortened.

Response: We have now added this information in the manuscript (line 300).

The discussion has 2 pages. In our opinion 2 pages of discussion is a usual length and all paragraphs of the discussion handle important aspects which are required for the topic and the results of our study.

Reviewer 2 Report

Comments and Suggestions for Authors

the authors have prepared an interesting work on the TLR-related pathway for ct-CD45 in the activation of dendritic cells. Contemporary discussion, results clearly presented, well discussed.

The manuscript is very good, I propose to expand on several aspects in the introduction/discussion
- TLR activation plays a protective role not only in the response to LPS (sepsis) but also (and this is more important in the response of vessels to agg-LDL, oxLDL, etc.). It is worth referring to this issue.
- Pharmacometric analysis of α1-adrenoceptor function in rat tail artery pretreated with lipopolysaccharides clearly indicates that the response is multiphase and clearly related to the function of the endothelium and endothelial nitric oxide synthases.
- the above answer modulates different types of contraction (from a pharmacological point of view) and the main "player" is TLR
- I wonder why the authors selected ILs (probably there were such possibilities) - why weren't IL-15, 18, 23 used?

from minor technical comments
- please correct the numbering in the references - it is duplicated.

Author Response

Thank you very much for your interest in our study and your constructive comments and feedback. We would like to address the comments and questions in the following way:

Reviewer: TLR activation plays a protective role not only in the response to LPS (sepsis) but also (and this is more important in the response of vessels to agg-LDL, oxLDL, etc.) It is worth referring to this issue.

Response: We agree that the protective or even inhibitory role of TLRs in certain diseases is maybe not referred enough in the original version of our paper. So, we have now added this information in our new version of our manuscript (line 58-62).

Reviewer: Pharmacometric analyses of a1-adrenoceptor function in rat tail artery pretreated with lipopolysaccharides clearly indicates that the response is multiphase and clearly related to the function of the endothelium and endothelial nitric oxide synthase – the above answer modulates different types of contraction (from a pharmacological point of view) and the main “player” is TLR

Response: Thank you for this input. We now mention this concept in the new version of our manuscript and cite appropriate new references (Ref. 12, 13) in order to support this statement (line 58-62).

Reviewer: I wonder why the authors selected ILs (probably there were such possibilities) – why weren´t IL-15, 18, 23 used

Response: You are right that someone can analyze and find many more cytokines in the supernatant of activated DCs. We have selected a panel of informative cytokines which are often used by immunologists because these factors are prominent modulators of adaptive immune responses. We have analyzed IL-23 and show the results in Figure 2.

Please correct the numbering in the references – it is duplicated

Response: Thank you for pointing to this mistake. We have now corrected the numbering.

Reviewer 3 Report

Comments and Suggestions for Authors

Lines 22, 180, 190, 276, : IFN- gamma is missing?

Line 41: inhibitor of t cell activation

Fig 1 ab: show in legend what colour coding means

Fig 1c: the c is overreaching the top left graph. Adapt the appearance of the MOCK to the graph pad design

Lines 100, 337, 345: Replace FACS with flow cytometry

Line 111: TNF- something is missing (alpha?)

Fig 2: Arrange the labelling in a uniform way. Either 45° or horizontal

Line 124: t-test: In my humble opinion an ANOVA should have been used here. Please discuss with a statistician.

Line 135: in vitro italics

Fig 4A/B: Better align the y axis labelling to overwrite the below labelling. Seriously, put some effort into it!

Lines 161, 174: no space needed between mean plus/minus SEM

Line 162: Use an ANOVA? Please discuss with a statistician.

Line 175: What post tests were used?

Fig 6B: replace g with a gamma

Line 191: What post tests were used?

Line 261: in vitro in vivo

Line 323: 1x10e5

Line 324: 1x10e4

Lines 334, 337: space after 50 is missing (use a safed-space, i.e. control-shift-space)

Author Response

Thank you very much for your interest in our study, your detailed feedback and constructive comments. We would like to address the comments and questions in the following way:

Reviewer: Lines 22, 180, 190, 276: IFN-gamma is missing?

Response: We are sorry for that error which obviously occurred during the formatting of the paper. You are right, IFN-g is missing. We have corrected it throughout the paper.

Line 41, inhibitor of T cell activation

Thank you for the hint, we have removed the “s” (line 41).

Fig 1 ab: show in legend what colour coding means

We now mention in the new version of our paper the colour coding of the presented histograms.

Fig 1c: the c is overreaching the lop left graph. Adapt appearance of the MOCK to the graph pad design

We have now corrected the design of Fig. 1c (line 101-103).

Lines 100. 337, 345: Replace FACS with flow cytometry

We have now replaced the requested changes.

Line 111: TNF-something is missing (alpha?)

You are right, TNF-a is missing. We have corrected it throughout the paper.

Fig 2: Arrange the labelling in a uniform way. Either 45° or horizontal

The labelling of our new Figure 2 is now uniform.

Line 124: t-test: in my humble opinion an ANOVA should have been used here. Please discuss with statistician

We have now followed your suggestion and analyzed the data with one way ANOVA with post hoc pairwise Tukey´s test. This is now also mentioned in the text line 125. The significance of the results remained like as it was presented in our first version of our manuscript.

Line 135: in vitro italics

This has been changed in the new version (line 137).

Fig 4A/B: Better align the y axis labelling to overwrite the below labelling. Seriously, put some effort into it!

We are sorry for this mistake and have now aligned the labelling of the y-axis in Figure 4A and B.

Line 161, 174: no space needed between mean plus/minus SEM

We have removed the spaces.

Line 162: Use an ANOVA? Please discuss with a statistician.

We have now followed your suggestion and analyzed the data with one way ANOVA with post hoc pairwise Tukey´s test. This is now also mentioned in the text line 163. The significance of the results remained like as it was presented in our first version of our manuscript.

Line 175: What post tests were used?

We have used one way ANOVA with post hoc pairwise Tukey´s test. This is now also mentioned in the text line 178.

Fig 6B: replace g with a gamma

We have now replaced g with g.

Line 191: What post tests were used?

We have used post hoc pairwise Tukey´s test. This is now also mentioned in the text, line 194.

Line 261: in vitro and in vivo

Both words are now presented in italic (line 264).

Line 323: 1x10e5

We are sorry for the mistake which obviously occurred during the transformation into the journal style. It is now corrected (line 327).

Line 324: 1x10e4

Is now corrected (line 328).

Line 334, 337: space after 50 is missing (use a safed-space, i.e. control-shift-space)

The space is now introduced.

Reviewer 4 Report

Comments and Suggestions for Authors

Paper by Stöckl  group  presents some evidences that ct-CD45 modulate functions of not matured dendritic cells interacting with TLR4. In particular modification of cytokine production profile has been described with increased production of IL-10 and a shift toward Th2 profile of T cells co-cultured with ct-CD45 stimulated CDs

Paper is well conceived and presents convincing experimental evidences. There are some minor comments that need to be addressed:  

In a previous paper the same group show that ct-CD45 inhibits T cells via TLR4 interaction whereas  TLR1, TLR2, and TLR6 appear not involved. Maybe a sets of experiments demonstrating that TLR1, TLR2, and TLR6  do not interact with ct-CD45 on DCs too would be useful.

In the discussion section, authors speculate on the different possible mechanisms and pathways for TLR4 signalling modulation by ct-CD45. In my opinion these mechanisms should be experimentally explored and dissected to better define the effect of TLR4/ct-CD45 interactions in DCs on intracellular signaling pathways leading to IL-10 production and accessory molecules (e.g. CD71, CD274, or CD209) surface expression

Author Response

Thank you very much for your interest in our study and your constructive comments and feedback. We would like to address the comments and questions in the following way:

Reviewer: Paper by Stöckl  group  presents some evidences that ct-CD45 modulate functions of not matured dendritic cells interacting with TLR4. In particular modification of cytokine production profile has been described with increased production of IL-10 and a shift toward Th2 profile of T cells co-cultured with ct-CD45 stimulated CDs

Paper is well conceived and presents convincing experimental evidences. There are some minor comments that need to be addressed:  

In a previous paper the same group show that ct-CD45 inhibits T cells via TLR4 interaction whereas  TLR1, TLR2, and TLR6 appear not involved. Maybe a sets of experiments demonstrating that TLR1, TLR2, and TLR6  do not interact with ct-CD45 on DCs too would be useful.

Response: We have previously shown (EJI, 2021, 51:3176) that ct-CD45 stimulates TLR4 expressing report cells but not TLR1, TLR2 or TLR6 expressing cells. Yet, we agree that this does not exclude the possibility that other pattern recognition receptors maybe involved on DCs when ct-CD45 is binding. We demonstrate, however, that ct-CD45 induced IL-10 and CD71 are almost completely inhibited when TLR4 is blocked. So, based on these data it is unlikely that additional receptor(s) are prominently involved.

Reviewer: In the discussion section, authors speculate on the different possible mechanisms and pathways for TLR4 signalling modulation by ct-CD45. In my opinion these mechanisms should be experimentally explored and dissected to better define the effect of TLR4/ct-CD45 interactions in DCs on intracellular signaling pathways leading to IL-10 production and accessory molecules (e.g. CD71, CD274, or CD209) surface expression

Response: We agree that molecular mechanisms induced upon TLR4/ct-CD45 interaction need to be experimentally explored in more detail. We have started to study the functional consequences in T cells where the interaction causes a striking and unexpected inhibition of T cell activation. This effect is not only detectable with ct-CD45 but also with an anti-TLR4 antibody. These studies are in an early phase and we hope to publish it in future papers.

Round 2

Reviewer 1 Report

Comments and Suggestions for Authors

The authors have revised the manuscript accordingly. I have no further comments.